# Spatiotemporal Evolution of Travel Pattern Using Smart Card Data

**Mu Lin [1], Zhengdong Huang [1,2,3,\*], Tianhong Zhao [2,3], Ying Zhang [2,3] and Heyi Wei [4]**

1 School of Urban Design, Wuhan University, Wuhan 430072, China; mulin@whu.edu.cn
2 Research Institute for Smart Cities, School of Architecture and Urban Planning, Shenzhen University, Shenzhen 518060, China; zhaotianhong2016@email.szu.edu.cn (T.Z.); y.zhang@szu.edu.cn (Y.Z.)
3 Guangdong-Hong Kong-Macau Joint Laboratory for Smart Cities & Guangdong Key Laboratory of Urban Informatics & Shenzhen Key Laboratory of Spatial Information Smart Sensing and Services, Shenzhen 518060, China
4 Geodesign Research Centre for Plant, Environment and Humans, Jiangxi Normal University, Nanchang 330022, China; weihy@whu.edu.cn
\* Correspondence: zdhuang@szu.edu.cn

**Abstract:** Automated fare collection (AFC) systems can provide tap-in and tap-out records of passengers, allowing us to conduct a comprehensive analysis of spatiotemporal patterns for urban mobility. These temporal and spatial patterns, especially those observed over long periods, provide a better understanding of urban transportation planning and community historical development. In this paper, we explored spatiotemporal evolution of travel patterns using the smart card data of subway traveling from 2011 to 2017 in Shenzhen. To this end, a Gaussian mixture model with expectation–maximization (EM) algorithm clusters the travel patterns according to the frequency characteristics of passengers' trips. In particular, we proposed the Pareto principle to negotiate diversified evaluation criteria on model parameters. Seven typical travel patterns are obtained using the proposed algorithm. Our findings highlighted that the proportion of each pattern remains relatively stable from 2011 to 2017, but the regular commuting passengers play an increasingly important position in the passenger flow. Additionally, focusing on the busiest commuting passengers, we depicted the spatial variations over years and identified the characters in different periods. Their cross-year usage of smart cards was finally examined to understand the migration of travel patterns over years. With reference to these methods and insights, transportation planners and policymakers can intuitively understand the historical variations of passengers' travel patterns, which lays the foundation for improving the service of the subway system.

**Keywords:** passenger clustering; smart cards; spatiotemporal analysis; Pareto front

## 1. Introduction

These days, thousands of modern cities have built the subway system to improve public transport efficiency and urban mobility. An in-depth understanding of urban mobility has a great contribution to the decision-making of transport management and urban planning. In order to explore urban mobility in detail, it is necessary to analyze personal travel behaviors from daily activities and monitor key indicators in future management. Traditional questionnaire-based travel surveys have been a common method for collecting information on individual travel behavior in previous studies, but this method is costly and limited in spatial and temporal resolution. With the development of information and communications technology (ICT) employed in the transport system, the availability of smart card data enables us to carry out an in-depth investigation of individual travel behaviors.

The smart card is designed for simplifying boarding or alighting transactions when passengers use the automated fare collection (AFC) systems in the public transport system. Consequently, the transaction data include the information about boarding time and

location, alighting time and location (depending on the charging system), and trip fee. The spatiotemporal information embedded in smart card data helps us create the individual trip trajectory. Compared to traditional survey data, smart card data has three advantages: first, the high resolution of spatial and temporal information embedded in smart card data makes it possible to identify personal travel behaviors on a finer scale; second, the large amount of travel data allow us to investigate the travel behaviors of the vast majority of passengers; and third, the convenience and low cost of data acquisition enable us to investigate the usage of the system over years.

Smart card data have been widely employed in urban transport studies. In view of previous works on exploring travel patterns, researchers primarily focused on either investigating the spatial and temporal characteristics of system usage or identifying the influential factors. There is an emerging body of research on investigating spatial and temporal characteristics, and some studies contribute to understanding the spatial and temporal variations of ridership [1–6], the evolution of urban spatial structure [7], jobs–housing relationship [8,9], and the interaction with built environment [10,11]. Moreover, the influence factors under various circumstances bring new insights for classic issues, such as route choice habit [12] and stickiness [13], public interest over new rail transportation [14], built environment assessment [15], climatic impact on travel intention [16], passenger flow under special events [17,18], congestion estimation [19] and migration of vulnerable groups [20,21]. These studies present valuable outcomes for the comprehension of travel patterns, however, they primarily focused on the short-term data of system usage, and only a few studies paid attention to the yearly usage of the system [8,22]. Employing the long-term smart card data is indispensable for drawing a complete picture of users and their usage, which lays the foundation for planners and policymakers to improve the system and further achieve sustainable urban mobility.

Various data mining techniques and algorithms have been developed to analyze the spatial and temporal patterns of residential travel. The K-means algorithm, C4.5 algorithm, Rough set-based algorithm, Naïve Bayes algorithm, K-NN algorithm, and Gaussian generative model have been applied for segmenting temporal characteristics [23–25]. Eigen decomposition is proposed as a transfer method from signal processing to extract common patterns, utilizing the principal component to compress the temporal feature appearance is dependent station variables [26]. With respect to the spatial features, the density-based spatial clustering of applications with noise (DBSCAN) algorithm is shown to be a feasible method to infer trip origins and destinations [27] and analyze the regularity of each cardholder [28]. Additionally, Nonnegative Matrix Factorization (NMF) and Hierarchical Ascendant Classification (HAC) have been used for identifying behavioral patterns [29] and estimating trip familiarity [30]. The above-mentioned works mainly focus on the classic clustering algorithms (such as K-means, DBSCAN, HAC, etc.) and matrix decomposition algorithms (such as NMF, Eigen decomposition, etc.). However, two issues still need to be considered in both two kinds of algorithms. First, data distribution and distance measurement criteria will significantly affect the result validity in classic clustering algorithms [31,32], while the influence grows in higher dimensional data. Second, the information loss always remains in the dimension reduction for matrix decomposition algorithms, which is hard but essential to explain the influence on the final result. Consequently, the algorithm for mining travel patterns should be less sensitive to data distribution and distance measurement criteria with information lossless.

The Gaussian mixture model has been widely used in various pattern analysis [33] and data aggregation scenarios [34–36]. As a statistical-based cluster method, the Gaussian mixture model is not only applied in diversified data distribution [37,38], but also calculated by Gaussian function parameters rather than the mutual distance among datasets. Furthermore, Eigen decomposition is not essential for input variables, indicating that the algorithm adapts to original feature vectors without information losses. In addition, a key parameter for unsupervised clustering is the number of clusters; however, many studies

determine the number of clusters based on a priori knowledge and lack a quantitative method to support it.

To fill this gap, this study is carried out to explore how users and their spatiotemporal travel behavior varied over a long period of time. Three key questions will be addressed: (1) how do users vary over a long period of time based on the unique smart card ID, (2) what are the spatiotemporal travel patterns of users, and (3) how to determine the best optimal number of clusters? To this end, we employed the Gaussian mixture model (GMM) and spatial analyses to examine the spatiotemporal characteristics of travel behaviors, using the smart card data of subway traveling from 2011 to 2017. Our contribution is summarized as follows:

- We built individual subway trip chains (i.e., the sequence of trips generated during the day, with the information of O-D times and locations) and explored individual travel patterns based on individual trip frequency.
- We proposed a user clustering scheme to unveil the distribution of trip frequency over the hour of the day for each user, employing the GMM with EM algorithm for clustering and integrated Pareto principle method to decide the number of clusters.
- We revealed the evolution of residents' personal travel patterns from 2011 to 2017, as well as the spatial and temporal distribution of each cluster.

The rest of the content is organized as follows. The cluster method and model parameters are explained in Section 2. Clustering results and the spatiotemporal characteristics of travel behaviors are presented and discussed in Section 3. Finally, Section 4 concludes the findings and presents our potential inferences.

## 2. Methods

### 2.1. Data Source and Preliminary Analysis

Smart card records were collected in Shenzhen, a modern city that serves as a window for China's reform and opening-up policy. In the last 40 years, Shenzhen has blossomed into one of the most important financial, manufacturing, and technological centers in China, attracting numerous migrants to work in this city. Luohu, Futian and Nanshan district formed the central urban areas, concentrating a large number of jobs in this city. The remaining districts and eastern suburbs formed peripheral urban areas. Due to the abundance of cheap residential areas in the peripheral urban area, a large amount of subway commuting occurs between the peripheral urban area and the central urban area. In 2004, Shenzhen built its first-ever subway line, i.e., the east portion of Line 1. To meet the needs for urban development, the subway system network in Shenzhen was established in 2011, which opened five subway lines (including Line 1 West, Line 2, Line 3, Line 4, and Line 5). The other three subway lines (Line 7, Line 9, and Line 11) had been built late in 2016 (Figure 1).

In this study, we collected the smart card records for the second or third week of every September from 2011 to 2017, avoiding the Chinese Mid-Autumn Festival and National Day holidays (the Chinese Mid-Autumn Festival is on 15 August by the Chinese lunar calendar, usually falling in September, and the National Day holidays start from 1 October). The entire dataset contains 155.72 million subway transactions and 18.94 million cardholders. Each transaction record includes the passenger card ID number, the time stamp of the transaction, the types of transactions (tap-in or tap-out from subway stations), the trip fee (only for the tap-out record), and the terminal equipment ID of transactions, and the subway station name.

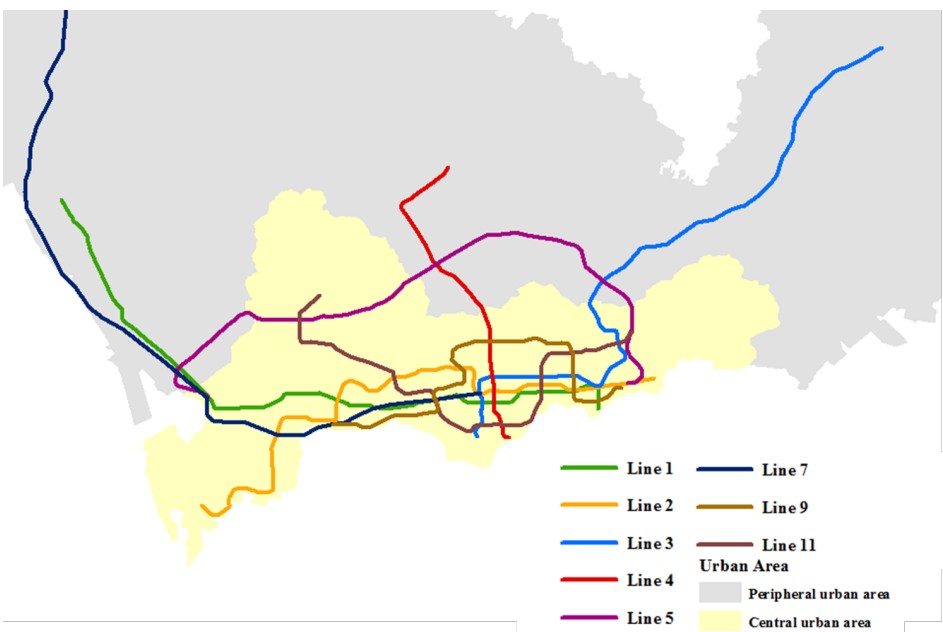

**Figure 1.** Spatial distribution of subway network in Shenzhen.

Compared with other attributes in smart card data, passenger (cardholder) IDs enable us to create a continuous timeline chart for the number of trips by specific users based on the seven-year dataset from 2011 to 2017, as shown in Figure 2. The horizontal axis indicates the year and the vertical axis indicates the number of cardholders, including new and continuous cardholders. Overall, from 2011 to 2017, cardholders increased from 1.7 million to 3.86 million. Between 2012 and 2017, the total number of continuous cardholders increased from 0.56 million to 1.87 million.

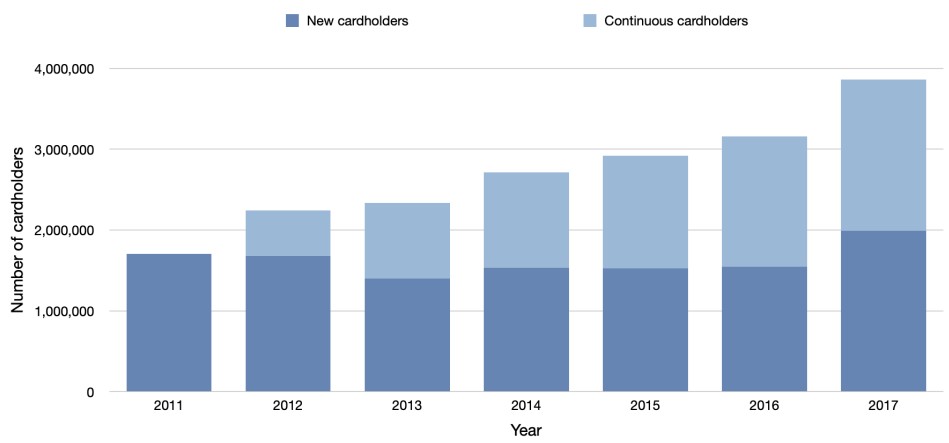

**Figure 2.** Composition of the number of cardholders from 2011 to 2017.

Travel time patterns can represent distinct groups of people's activity characteristics [6]. Figure 3 describes the distribution of the proportion of trips over the day of the week each year. No particular distinction can be detected from the distribution, except that the proportion of trips that occurred on workdays continued to grow slightly over the years.

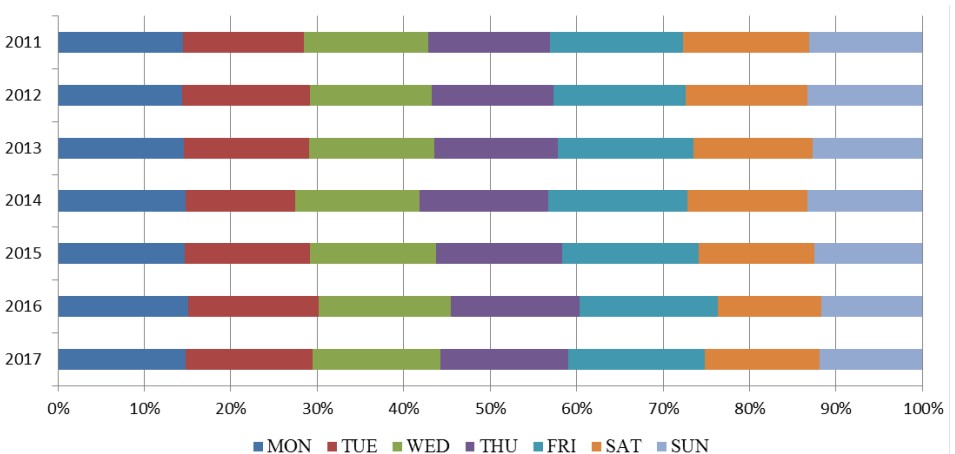

**Figure 3.** The proportion of trips over the day of the week (2011–2017).

### 2.2. Vector of Individual Trip Features

Trip frequency, commuting time, and commuting distance were usually taken as clustering indicators to explore travel patterns based on different research purposes. Unlike commuting time and distance, which mainly contribute to identifying temporal characteristics of personal movement, the trip frequency can uncover passengers' activity intensities and their preferences for using public transit systems.

To examine the travel behaviors of individual passengers in detail, we used a 168-dimension (7 × 24 h per week) vector to compute the trip frequency of each passenger, thus each element in the vector represents the number of trips during the hour corresponding to the element index as follow:

$$A_i = (a_{i1}, a_{i2}, a_{i3}, \cdots, a_{iN})$$ (1)

where $A$ is the trip frequency per hour, $i$ is the sequential number of cardholders and $N$ is the dimension (168 in this case).

### 2.3. Gaussian Mixture Model

Inspired by smart card research reviews [22,39,40], any multidimensional data [41–43] can be fitted by the Gaussian mixture model as the further extension of previous research, indicating that the cardholders' transit records could be considered as a kind of mixture model that consists of several components. Each component can be regarded approximately following the Gaussian distribution, which is also conditionally independent between any two clusters. The two essential conditions lay the foundation of the Gaussian mixture model clustering, leading to a general acceptance of the explanation. The Gaussian mixture models [44] are given by:

$$P(y \mid \theta) = \sum_{k=1}^{K} \alpha_k \phi(y \mid \theta_k)$$ (2)

where $K$ is the number of Gaussian functions in the model capturing the variety of trip features and $\alpha_k$ is the coefficient of the $K_{th}$ component, which usually represents the proportion of different types of clusters. The individual trip can be generated from the $K_{th}$ component. Meanwhile, $\phi(y \mid \theta_k)$ denotes the probability density function of the Gaussian distribution as follows:

$$\phi(y \mid \theta_k) = \frac{1}{\sqrt{(2\pi)}\sigma_k} \exp\left(-\frac{(y - \mu_k)^2}{2\sigma_k^2}\right)$$

$$\theta_k = \left(\mu_k, \sigma_k^2\right)$$ (3)

For each Gaussian component, $\mu_k$ represents the average vector of trip frequency in the $k_{th}$ component and $\sigma_k$ is the standard deviation for the $k_{th}$ Gaussian distribution. The selection of the $k_{th}$ component depends on the probability $\alpha_k$ corresponding to the proportion of each cluster in this analysis.

*2.4. Expectation-Maximization Algorithm*

We solve the Gaussian mixture model with the expectation-maximization algorithm and obtain different clusters based on the trip frequency features. In theory, for a solution of the Gaussian mixture model, the parameter estimation method should be adopted to identify the variables $\alpha_k, \mu_k,$ and $\sigma_k$. However, the mixture distribution of the sample assumes that the category of the temporal vector is unknown. In order to solve the problem, we introduce a latent variable $\gamma_{jk}$ describing the responsivity of the component to individual trip $y_j$. Therefore, the dataset can be expanded as follows:

$$\left(y_j, \gamma_{j1}, \gamma_{j2}, \cdots, \gamma_{jk}\right), j = 1, 2, \cdots, N \tag{4}$$

From the expanded data, the likelihood function for forming the passenger clusters is given by:

$$P(y, \gamma \mid \theta) = \prod_{j=1}^{N} P\left(y_j, \gamma_{j1}, \gamma_{j2}, \cdots, \gamma_{jk}\right) \tag{5}$$

The equation can be also written as:

$$P = \prod_{k=1}^{K} \alpha_k^{nk} \prod_{j=1}^{N} \left[\frac{1}{\sqrt{(2\pi)}\sigma_k} \exp\left(-\frac{(y-\mu_k)^2}{2\sigma_k^2}\right)\right]^{\gamma_{kk}} \tag{6}$$

where the variables are given as:

$$n_k = \sum_{j=1}^{N} \gamma_{jk}$$
$$N = \sum_{k=1}^{K} nk \tag{7}$$

In terms of the likelihood function, the EM algorithm takes the E-step and the M-step to interpret unknown parameters in an iterative solution. The E-step of the algorithm involves calculating the expectation of the likelihood function to identify the maximization probability of classification. The expectation function can be written in a logarithmic form as:

$$Q\left(\theta, \theta^{(i)}\right) = E\left[\log P(y, \gamma \mid \theta)y, \theta^{(i)}\right]$$
$$= \sum_{k=1}^{K}\left\{\sum_{j=1}^{N}\left(E_{\gamma_{jk}}\right)\log\alpha_k + \sum_{j=1}^{N}\left(E_{\gamma_{jk}}\right)\left[\log\left(\frac{1}{\sqrt{2\pi}}\right) - \log\alpha_k - \frac{1}{\sqrt{2\pi}}(y_j - \mu_k)^2\right]\right\} \tag{8}$$

where $E_{\gamma_{jk}}$ is the parameter estimation for the weightiness of clusters, also meaning the cluster membership of each component, computed as

$$\hat{\gamma}_{jk} = E_{\gamma_{jk}} = \frac{\alpha_k \phi\left(y_j \mid \theta_k\right)}{\sum_{k=1}^{K} \alpha_k \phi\left(y_j \mid \theta_k\right)} \tag{9}$$

Moreover, the initial variable value should be incorporated into the function to update the further iterative variable in this anticipation function. The iterative method in the M-step uses the derivation of this anticipation function to compute the maximum value for new iterative variable values, which can be specified by:

$$\mu^{i+1}, \sigma^{i+1}, \alpha^{i+1} = \arg\max Q\left(\mu, \sigma, \alpha, \mu^i, \sigma^i, \alpha^i\right) \tag{10}$$

where the values can be estimated in their partial derivatives, as follows:

$$\hat{\mu}_k = \frac{\sum_{j=1}^{N} \hat{\gamma}_{jk} y_i}{\sum_{j=1}^{N} \hat{\gamma}_{jk}}, k = 1,2,3,\cdots,K$$

$$\hat{\sigma}_k^2 = \frac{\sum_{j=1}^{N} \hat{\gamma}_{jk}\left(y_j - \mu_k\right)^2}{\sum_{j=1}^{N} \hat{\gamma}_{jk}}, k = 1,2,3,\cdots,K \tag{11}$$

$$\hat{\alpha}_k = \frac{\sum_{j=1}^{N} \hat{\gamma}_{jk}}{N}, k = 1,2,3,\cdots,K$$

The iterative calculation in the E-step and M-step continues until the model converges. The workflow for GMM clustering is presented in Figure 4. According to the final result, each trip vector could be trained to test the corresponding probability for all cluster centers and finally allocated to the appropriate cluster.

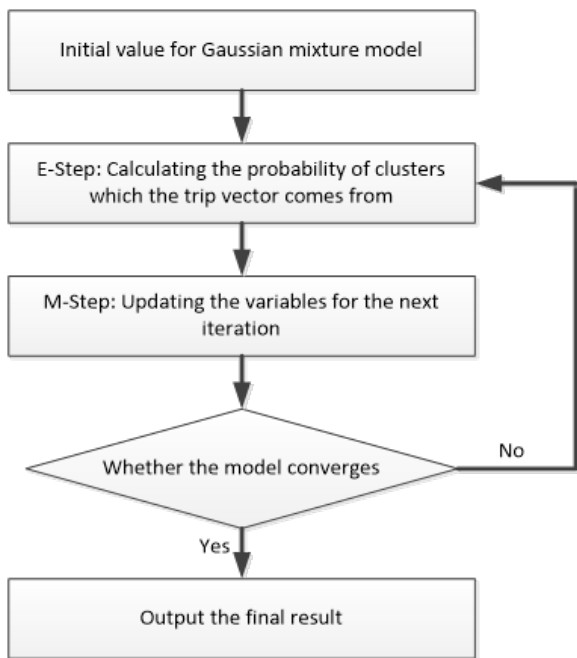

**Figure 4.** Workflow for GMM algorithm.

*2.5. Parameter Choice*

Unlike the parameters estimated by the algorithm $(\mu, \sigma, \alpha)$, the number of clusters should be selected beforehand. In this section, we introduced multiple criteria to evaluate the performance of distinct cluster numbers. Due to the complexity of the distribution of clustering samples, it is significant that clustering results should keep more original sample information. Considering to minimize the information losses in clustering, Akaike information criterion (AIC) is adopted for evaluating the volume of information entropy [45], which is denoted as:

$$AIC = 2m - 2\ln(P) \tag{12}$$

where $m$ represents the number of model parameters and $P$ refers to the likelihood function value. Generally, a lower AIC index reflects more abundant information entropy in the clustering solution.

Another consideration for unsupervised clustering is to emphasize cluster validity. CalinskiHarabaz (CH) criterion can be applied to assess the quality of clusters with specified cluster numbers when ground truth labels are not known. CalinskiHarabaz index measures

the weights between the cohesion of the intra-class average distance and the separation of the inner-class distance, given as:

$$\text{CH} = \frac{\text{Tr}(B_k)}{\text{Tr}(W_k)} \times \frac{N - k}{k - 1} \tag{13}$$

where $k$ is the number of clusters and $N$ is the number of individual trips. $\text{Tr}(B_k)$ and $\text{Tr}(W_k)$ represent the trace of the inter-class dispersion matrix and the intra-class dispersion matrix. According to the definition of CH index, a higher CH index indicates a better-defined cluster number with reasonable allocation on intra-class distance and inter-class distance.

However, in the majority of cases, the performance of the same cluster number on these two evaluations may be self-contradiction, because more clusters always enhance the information expression and reduce the sensitivity among cluster features. To balance the evaluation indexes of the two methods, we introduced the Pareto principle to extract the solution set in a multi-objective decision [46]. For the multi-objective decision of parameter choice, two objective functions are organized as $f^*(x)$ for minimization over cluster number solution $x$, shown as:

$$\min f^*(x) = \{f_1(x), f_2(x)\} \tag{14}$$

while the Pareto-optimal solution $\min f^*(x)$ is selected for the reason that it is impossible to get better $f_i(x)$ without negative effect on $f_j(x)$, given the original situation. We attempt to pick all appropriate clustering numbers that satisfied the Pareto principle, forming the Pareto front related to clustering performance as Figure 5.

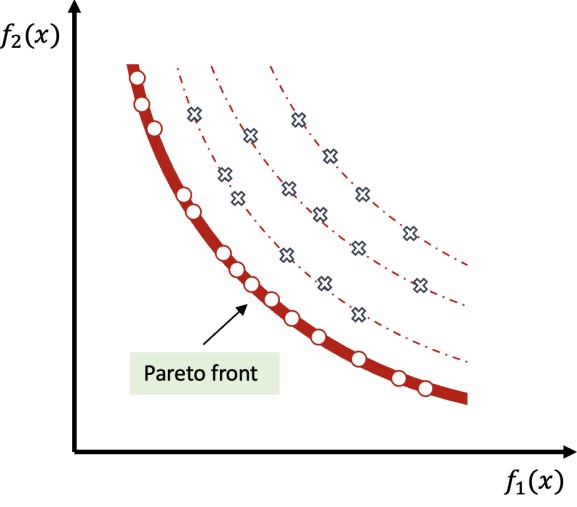

**Figure 5.** The concept of Pareto front through the usage of multiple objective functions.

## 3. Results and Discussion

### 3.1. Clustering Results of Gaussian Mixture Model

Prior to presenting the clustering results, we present and discuss the determination of model parameters, clustering results of GMM, and spatiotemporal characteristics of entire passengers and continuous cardholders. Figure 6 demonstrates clustering performance varying the cluster number from 2 to 30 while the horizontal coordinate is the normalization value of $\text{CH}^{-1}$ and the vertical coordinate is the normalization value of AIC (the results with the lowest performance on every single criterion are excluded in our choices). In total, thirteen Pareto-optimal solutions on cluster number have been identified.

To identify the exact clustering number of the Pareto frontier, we applied the dominant principle to select the best Pareto solution, which is donated as:

$$DomiScore(x) = \sqrt{\sum_{i=1}^{n} \left| \frac{f_i(x) - \max f_i(x)}{\max f_i(x) - \min f_i(x)} \right|^2} \tag{15}$$

while max $f_i(x)$ and min $f_i(x)$ represent the highest and lowest performance in the $i$th criterion. According to the comprehensive performance in both criteria, we determine the number of clusters $k$ as 7 from the Pareto frontier.

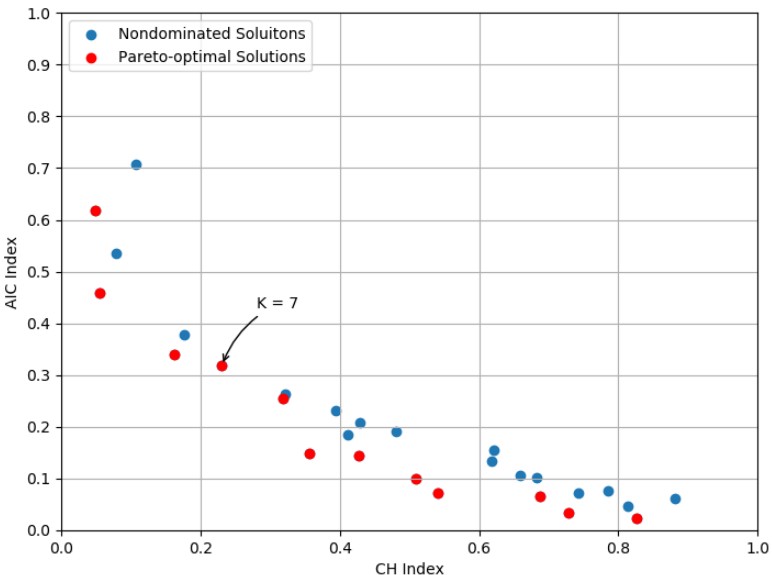

**Figure 6.** Pareto front for the best clustering performance.

As shown in Figure 7, seven clusters were generated by the GMM, indicating the representative travel behaviors of passengers. In view of total trip frequency, clusters 1, 2, 3, and 4 present much active traveling status among all types of clusters. Cluster 5 has some non-peak travelers on the first four days of the week. Cluster 6 has few subway travel trips during weekdays, but more active trips on the weekends. There is also a distinct group (cluster 7) who seldom commutes during weekdays but travel frequently on Friday.

Peak features are characterized by obvious distinctions among clusters. Clusters 1, 3, and 4 follow the regular two-peak pattern, these cardholders should be commuters. Specifically, the morning peak in cluster 1 (from 6:00 a.m.) is at least one hour earlier than in other clusters. However, the evening peak in these three clusters extends from 6:00 p.m. to 8:00 p.m., which is in line with the normal commute time for most office workers. Cardholders in cluster 3 have relatively consistent working hours, as they only travel during morning and evening peak hours and rarely travel during other hours. Unlike cluster 3, cluster 4 is not concentrated in the morning and evening peak hours, and they have more flexible working hours. In addition, cluster 2 displays the inconspicuous three-peak pattern for the so-called night owls, indicating that another evening peak occurs around 10:00 p.m. The travel activities of cluster 6 in a week are mainly concentrated on Fridays and Saturdays. This cluster should be the students living in schools. Since they live in school on weekdays and do not need to travel, they may need to travel for social activities on weekends. Cluster 7 is unique among all clusters in that it has a relatively regular but inactive travel frequency from Monday to Thursday, with the main trips concentrated in the evening peak. However, the daily trips on Friday seem to be activated, while trips on weekends nearly fall into sleep. Based on their travel pattern, we can infer that these people may be part of the passengers who choose other modes of transportation (bus, taxi) from Monday to Thursday in the morning rush hour due to traffic congestion or other factors, and occasionally choose the subway in the evening instead. Considering the abnormally active trip frequency on Friday, they may choose the subway to go to more important destinations such as airports, rail stations or ports based on the belief in the reliability of the rapid transit system. This speculation can also explain why this kind of passenger becomes less active on weekends.

As for the comparison of trip frequency between weekdays and weekends, passengers in clusters 2, 3, 4, 5, and 7 take the subway less frequently on the weekends than on weekdays, whereas passengers in cluster 1 follow the same timetable for both workdays and weekends. This implies that those passengers have a fixed routine seven days a week. Moreover, the passengers of cluster 6, the largest user group (23.8%), prefer to travel in the afternoon or evening at weekends, but their working day usage suggests that they probably do not take the subway for commuting.

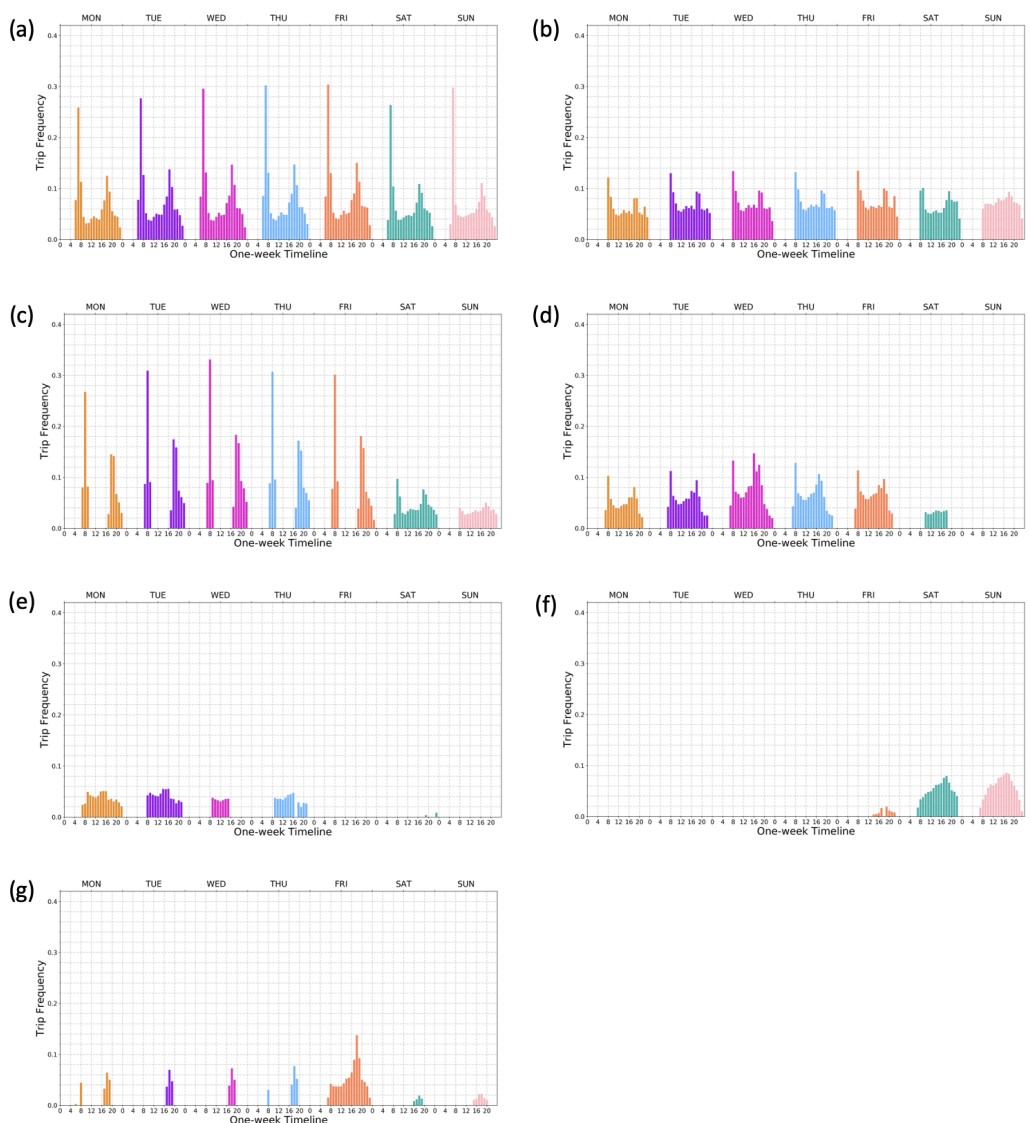

**Figure 7.** The configurations of travel behaviors in clusters 1–7. (**a**) Temporal profile of cluster 1: 549,327 passengers (2.9%); (**b**) temporal profile of cluster 2: 2,239,660 passengers (11.8%); (**c**) temporal profile of cluster 3: 3,798,313 passengers (20.0%); (**d**) temporal profile of cluster 4: 2,234,894 passengers (11.8%); (**e**) temporal profile of cluster 5: 2,825,129 passengers (14.9%); (**f**) temporal profile of cluster 6: 4,521,905 passengers (23.8%); and (**g**) temporal profile of cluster 7: 2,772,880 passengers (14.6%).

## 3.2. Passenger Structures and Travel Characteristics

We statistically analyze the structure and travel characteristics of each cluster based on the clustering results from the previous section. For a more comprehensive evaluation of the variations in travel behaviors over the years, we examined the distribution of the proportion of passengers across seven clusters every year, as shown in Figure 8. Although the

proportions of clusters in the sequence slightly fluctuate over the years, the passenger structure remains relatively stable in the entire dataset. Cluster 6 makes up at least 20% of passenger records, ranking first since 2011. However, the detailed temporal profile of cluster results shows that cluster 3 is approaching the occupancy rate of cluster 6, meaning that an increasing proportion of typical bimodal passengers select the subway as their commuting choice. In addition, for the representative commuting clusters, clusters 1 and 2 show a growth trend, suggesting that high frequency travelers also have more trust in the urban subway system.

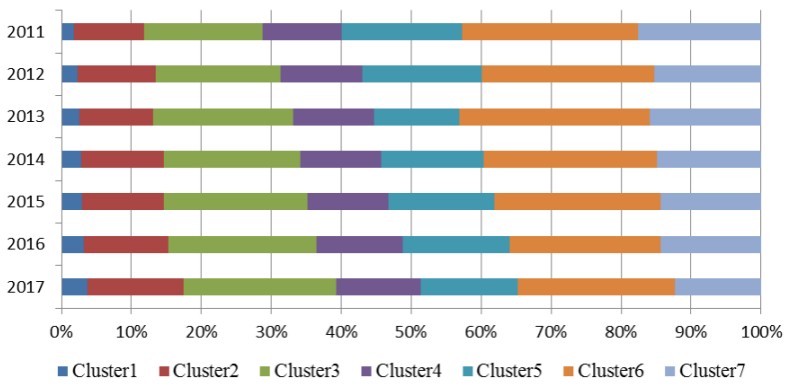

**Figure 8.** The proportion of passengers over clusters (2011–2017).

Figure 9 presents the average travel time of passengers belonging to each cluster every year. Obviously, the travel time maintains the same trend each year for all clusters with the fact that regular passengers (clusters 1–4) cost more time over years. Particularly, the time cost of cluster 1 in 2017 increased by 100 seconds over the 2011 level. Considering the general subway speed in Shenzhen (about 50–70 km/h), passengers in cluster 1 may be less sensitive to commuting time.

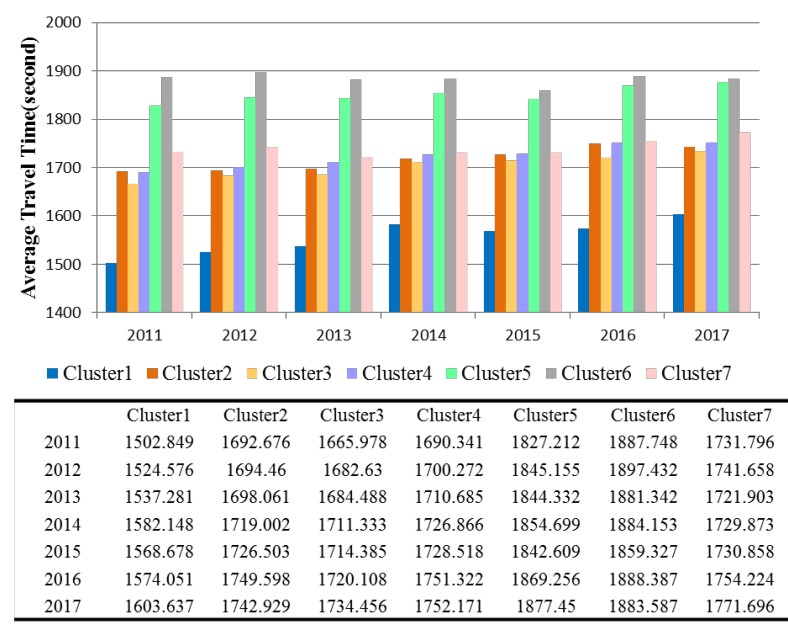

|      | Cluster1 | Cluster2 | Cluster3 | Cluster4 | Cluster5 | Cluster6 | Cluster7 |
|------|----------|----------|----------|----------|----------|----------|----------|
| 2011 | 1502.849 | 1692.676 | 1665.978 | 1690.341 | 1827.212 | 1887.748 | 1731.796 |
| 2012 | 1524.576 | 1694.46  | 1682.63  | 1700.272 | 1845.155 | 1897.432 | 1741.658 |
| 2013 | 1537.281 | 1698.061 | 1684.488 | 1710.685 | 1844.332 | 1881.342 | 1721.903 |
| 2014 | 1582.148 | 1719.002 | 1711.333 | 1726.866 | 1854.699 | 1884.153 | 1729.873 |
| 2015 | 1568.678 | 1726.503 | 1714.385 | 1728.518 | 1842.609 | 1859.327 | 1730.858 |
| 2016 | 1574.051 | 1749.598 | 1720.108 | 1751.322 | 1869.256 | 1888.387 | 1754.224 |
| 2017 | 1603.637 | 1742.929 | 1734.456 | 1752.171 | 1877.45  | 1883.587 | 1771.696 |

**Figure 9.** The average travel time of passengers in each cluster (2011–2017).

### 3.3. Spatio-Temporal Evolution of Cluster

Analyzing the temporal and spatial evolution of various passenger clusters might assist in exposing the evolving laws of urban spatial structure [8]. We illustrate the spatiotemporal variation of different clusters from the spatial distribution changes for one cluster and the transfer between clusters. Figure 10 describes the spatial distribution of

travel patterns of cluster 1, presenting the classified boarding stations for cluster 1. For the spatial variations, two stages can be identified from 2011 to 2017: (1) germination development stage (2011–2014); and (2) axial growth stage (2014–2017). The first stage indicates that stations' ridership increased in both peripheral and central urban areas in relative terms and synchronous steps. In 2014, the daily amount of travelers at three stations exceeded 5000. The second stage witnessed fast-growing subway patronage since 2015. By 2017, the number of top-level stations with a large number of travelers increased to sixteen. Interestingly, ridership growth has mainly concentrated on stations along with Line 1 and Line 4 in the second stage. In fact, large-scale urban development and urban renewal have led to land use restructuring, densification, and gentrification, which may partially explain the increase in subway traveling.

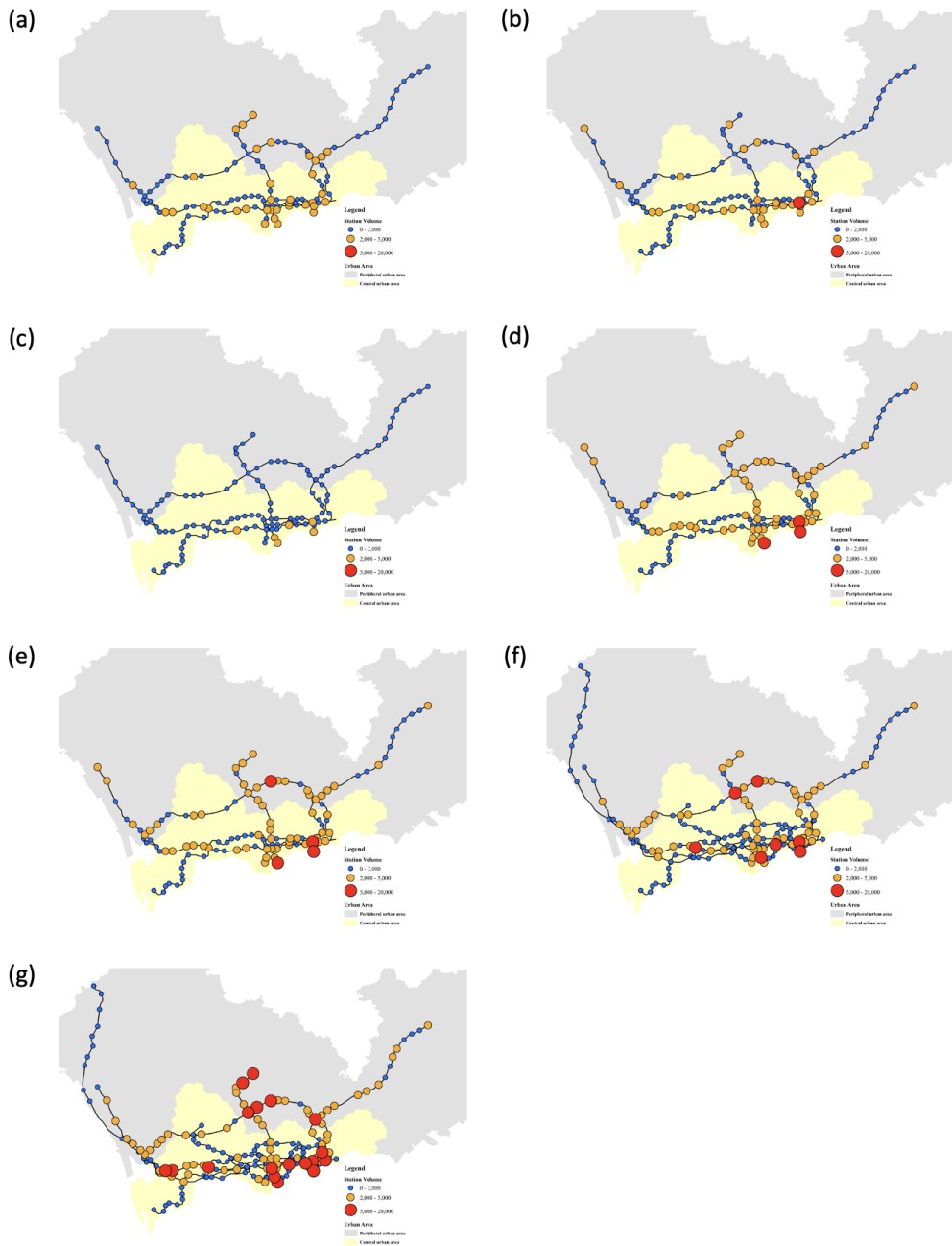

**Figure 10.** Spatial distribution of daily station volume for cluster 1 (2011–2017). (**a**) cluster 1 in 2011; (**b**) cluster 1 in 2012; (**c**) cluster 1 in 2013; (**d**) cluster 1 in 2014; (**e**) cluster 1 in 2015; (**f**) cluster 1 in 2016; and (**g**) cluster 1 in 2017.

Information on each passenger's ID is embedded in smart card data, therefore we can find out which cluster each passenger belongs to each year. As a result, the cluster migration of each individual passenger can be tracked over years. Figure 11 delineates the cluster transition matrix between 2011 and 2017. In the Sankey diagram, the order of clusters on both sides is ranked by the proportion of passengers in the respective years.

For continuous cardholders, there were significant changes in passengers' travel behaviors, i.e., only one-third of passengers stayed in the same cluster after six years. Moreover, the proportion of commuting passengers increased in 2017, implying that passengers with tidal characteristics (clusters 1–4) have become the main force of continuous cardholders. In addition, a larger proportion of passengers switched from cluster 1 (traveling most frequently) in 2011 to other clusters (traveling less frequently) in 2017, implying that some passengers have started a slow lifestyle or they have changed their travel modes.

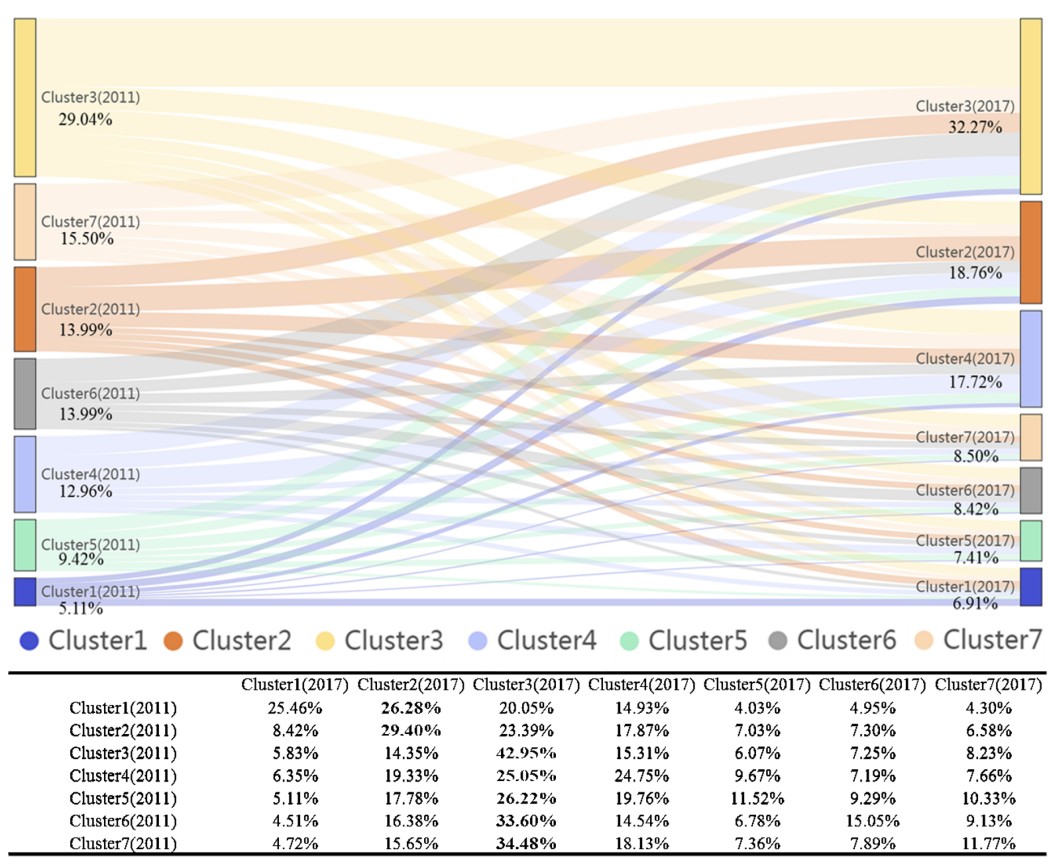

| | Cluster1(2017) | Cluster2(2017) | Cluster3(2017) | Cluster4(2017) | Cluster5(2017) | Cluster6(2017) | Cluster7(2017) |
|---|---|---|---|---|---|---|---|
| Cluster1(2011) | 25.46% | 26.28% | 20.05% | 14.93% | 4.03% | 4.95% | 4.30% |
| Cluster2(2011) | 8.42% | 29.40% | 23.39% | 17.87% | 7.03% | 7.30% | 6.58% |
| Cluster3(2011) | 5.83% | 14.35% | 42.95% | 15.31% | 6.07% | 7.25% | 8.23% |
| Cluster4(2011) | 6.35% | 19.33% | 25.05% | 24.75% | 9.67% | 7.19% | 7.66% |
| Cluster5(2011) | 5.11% | 17.78% | 26.22% | 19.76% | 11.52% | 9.29% | 10.33% |
| Cluster6(2011) | 4.51% | 16.38% | 33.60% | 14.54% | 6.78% | 15.05% | 9.13% |
| Cluster7(2011) | 4.72% | 15.65% | 34.48% | 18.13% | 7.36% | 7.89% | 11.77% |

**Figure 11.** Clusters migration between 2011 and 2017.

## 4. Conclusions

Smart card data have been widely applied in transport studies to uncover human travel behaviors. However, few studies have been conducted to explore the dynamics of travel behaviors over years. To fill this research gap, this study was conducted to understand spatiotemporal characteristics of human travel behaviors by examining the variations of passengers and their travel behaviors, using the smart card data of subway traveling from 2011 to 2017. To this end, a Gaussian mixture model was employed to examine the spatiotemporal patterns of passengers' travel behaviors. In particular, we propose the Pareto frontier method to determine the number of clusters, which is more reasonable than the traditional empirical-based method. Moreover, the dynamic changes of continuous cardholders and their travel behaviors over years were examined as well.

We found that no significant difference in the distribution of the proportion of trips over the day of the week can be identified between years. However, seven clusters were generated by the Gaussian mixture model based on trip frequency, indicating distinct

travel patterns over the hour of the day and between weekdays and weekends. Moreover, the proportion of passengers in each cluster varied significantly over years, showing that the proportion of commuting passengers increased year by year. In addition, for the spatial variations of travel patterns of cluster 1, two stages can be identified from 2011 to 2017, i.e., germination development stage (2011–2014) and the axial growth stage (2014–2017). For continuous cardholders, significant changes in passengers' travel behaviors were highlighted between 2011 and 2017, indicating that only one-third of passengers stayed in the same cluster after six years. Moreover, compared to 2011, commuters have become the main force of continuous cardholders in 2017. In addition, around 70% of passengers have switched from cluster 1 (traveling most frequently in 2011) to other clusters (traveling less frequently) in 2017, implying that some passengers have started a slow lifestyle or they have changed their travel modes.

This study has several limitations. We mainly focused on uncovering the spatiotemporal dynamics of passengers and their travel behaviors over a long period of time, but neglect the reasons (travel purposes) behind travel behaviors. The travel purposes of individual passengers are complex and might be correlated with the distribution of land use patterns and urban facilities as well as personal habits and preferences, which are beyond the scope of this study. We only used on-week data for each year due to data unavailability, and collecting more data might uncover more comprehensive portraits of human travel behaviors. However, we believe that our study still provides useful information and knowledge on the spatiotemporal dynamics of passengers and their travel behaviors in the long term. With reference to these methods and insights, other researchers can explore the long-term dynamics of individual travel behaviors in detail. Furthermore, these findings lay the foundation for transportation planners and policymakers to better understand and further improve the service of the subway system.

**Author Contributions:** Conceptualization, M.L.; methodology, M.L. and Z.H.; validation, M.L., Y.Z. and T.Z.; data curation, M.L. and Z.H.; writing—original draft preparation, M.L., T.Z. and Y.Z.; writing—review and editing, M.L., Z.H., T.Z., Y.Z. and H.W. All authors have read and agreed to the published version of the manuscript.

**Funding:** This research was funded by the National Natural Science Foundation of China (NSFC) [Number: 42071357, 41901389, 71961137003], Guangdong Science and Technology Strategic Innovation Fund (the Guangdong-Hong Kong-Macau Joint Laboratory Program), [Number: 2020B1212030009], and Shenzhen Key Laboratory of Digital Twin Technologies for Cities [Number: ZDSYS20210623101800001].

**Institutional Review Board Statement:** Not applicable.

**Informed Consent Statement:** Not applicable.

**Data Availability Statement:** The data presented in this study are available on request from the corresponding author.

**Conflicts of Interest:** The authors declare no conflict of interest.

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
