# Peer review of "Spatiotemporal Evolution of Travel Pattern Using Smart Card Data"

_sustainability, doi:10.3390/su14159564_

Round 1

Reviewer 1 Report

This paper explores the temporal and spatial evolution of travel patterns using smart card data in the Shenzhen Metro from 2011 to May 2017. And a Gaussian mixture model with EM algorithm and Pareto parameter choice was employed to uncover the spatiotemporal patterns of passengers’ travel behaviors.

The paper is well organized and its presentation is good. However, some issues still need to be improved:

1. The definition of " continuous usage " passenger IDs from 2011 to 2017 in the data source is not clear enough, and the number distribution chart is unreadable enough.

2. The 7th cluster should be explained in more detail to enhance the credibility.

3. the contrast experiment should be conducted to illustrate the reliability of the results.

4. There are some mistakes in the paper, such as the figure 6 in line 225, please check it carefully.

Reviewer 2 Report

The authors analyse six years’ smartcard data in Shenzhen using a Gaussian mixture model and identify seven clusters with distinct travel patterns. These were analysed over the years to understand travel behaviour change. The paper is well drafted, technically sound and useful for transport planners.

However, a concern is regarding the originality of the methodology. Using Gaussian mixture model for analysing smart card data has been attempted before (eg. Lee et al. 2019). The specific method that the authors use – that of using Gaussian mixture models for clustering public transport trips based on travel behaviour, and analysing their year-to-year changes – also exists in the literature (see Briand et al. 2016 and 2017). The authors have sited the latter work but not the former. It is not clear how the methodology presented in this paper improves the existing methods. The authors may consider detailing the improvement, without which the paper fails to attract research value, and limits itself as an implementation report of an existing methodology in another city.

References

Lee, E. H., Lee, I., Cho, S. H., Kho, S. Y., & Kim, D. K. (2019). A travel behavior-based skip-stop strategy considering train choice behaviors based on smartcard data. Sustainability11(10), 2791.

Briand, A. S., Côme, E., Trépanier, M., & Oukhellou, L. (2017). Analyzing year-to-year changes in public transport passenger behaviour using smart card data. Transportation Research Part C: Emerging Technologies79, 274-289.

Briand, A. S., Côme, E., El Mahrsi, M. K., & Oukhellou, L. (2016). A mixture model clustering approach for temporal passenger pattern characterization in public transport. International Journal of Data Science and Analytics1(1), 37-50.

Reviewer 3 Report

The widespread use of cards for travel in public transportation means generate valuable data about users and the corresponding load on transportation systems.

The authors analyze the long-term spatiotemporal patterns of urban mobility and spatiotemporal evolution of travel patterns using the smart card data of underground (metro) use in Shenzhen (China) from 2011 to 2017. They selected travel patterns and time of the years to avoid perturbations brought by particular events, like holidays or other external factors.

A Gaussian Mixture Model and spatial analyses are used to examine the spatiotemporal characteristics of travel behaviors for metro users with or without subscription cards. Findings are certainly useful for planning purposes for public transportation administration.

I agree with the authors that other researchers can explore the long-term dynamics of individual travel behaviors in detail. Even more, I consider that the methodology proposed can be extended to various public transportation means using cards (railways, buses, waterways, etc.)

Round 2

Reviewer 1 Report

The paper has been revised according to the previous comments. I think it is suitable to publish in journals in its current form.

Reviewer 2 Report

The article can be  published in its current form now.